# Efficacy of Kan Jang^®^ in Patients with Mild COVID-19: A Randomized, Quadruple-Blind, Placebo-Controlled Trial

**DOI:** 10.3390/ph16091196

**Published:** 2023-08-22

**Authors:** Levan Ratiani, Elene Pachkoria, Nato Mamageishvili, Ramaz Shengelia, Areg Hovhannisyan, Alexander Panossian

**Affiliations:** 1Department of Infectious Diseases, The First University Clinic, Tbilisi State Medical University, Gudamakari St., Tbilisi 0141, Georgia; l.ratiani@tmsu.edu (L.R.); e.pachkoria@tsmu.edu (E.P.); 2Department for History of Medicine and Bioethics, Faculty of Medicine, Tbilisi State Medical University, Vazha-Pshavela Ave. 33, Tbilisi 0162, Georgia; nato15sg@yahoo.com (N.M.); r.shengelia@tmsu.edu (R.S.); 3Institute of Fine Organic Chemistry of the National Academy of Science, Azatutian Ave. 26, Yerevan 375014, Armenia; dopingareg@gmail.com; 4Phytomed AB, Sjöstadsvägen 6A, Lgh 1004, 59344 Västervik , Sweden

**Keywords:** adaptogens, Kan Jang^®^, *Andrographis paniculata*, *Eleutherococcus senticosus*, clinical trial, mild COVID-19, IL-6, inflammatory symptoms

## Abstract

***Background and aim***. This study aimed to assess the efficacy of the treatment of Kan Jang^®^, a fixed combination of *Andrographis paniculata* (Burm. F.) Wall. ex. Nees and *Eleutherococcus senticosus* (Rupr. & Maxim.) Maxim extracts in patients with mild symptoms of COVID-19. ***Methods***. One hundred and forty patients received six capsules of Kan Jang^®^ (*n* = 68, daily dose of andrographolides—90 mg) or placebo (*n* = 72) and supportive treatment (paracetamol) for 14 consecutive days in a randomized, quadruple-blinded, placebo-controlled, two-parallel-group design. The efficacy outcomes were the rate of cases turning to severe, the detection rate of coronavirus SARS-CoV-2 over the time of treatment, the duration, and the severity of symptoms (sore throat, runny nose, cough, headache, fatigue, loss of smell, taste, pain in muscles) in the acute phase of the disease. Other efficacy measures included improving cognitive and physical performance, quality of life, and the levels of inflammatory blood markers—interleukin 6 (IL-6), C-reactive protein, and D-dimer. ***Results***. Kan Jang^®^ significantly (*p* < 0.05) reduced the rate of cases turning to severe (5.36%) compared to the placebo (17.86%) and decreased the detection rate of SARS-CoV-2 virus over the time of the treatment. The statistical difference in the rates of patients with clinical deterioration in the Kan Jang treatment and placebo control groups was significant (*p* = 0.0176) both in the 112 patients in the included-per-protocol (IPP) analysis and in the 140 patients in the intended-to-treat (ITT) analysis (*p* = 0.0236); the absolute risk reduction in cases thanks to the Kan Jang treatment was 12.5%, and the number we needed to treat with Kan Jang was 8. The patient’s recovery time (number of sick days at the home/clinic) was shorter in the Kan Jang group compared with the placebo group. The rate of attenuation of inflammatory symptoms in the Kan Jang^®^ group was significantly higher, decreasing the severity of cough, sore throat/pain, runny nose, and muscle soreness compared with the placebo group. Kan Jang^®^ significantly decreased the Wisconsin Upper Respiratory Symptoms scores compared to the placebo in the sample size of 140 patients. However, the relief of fatigue and headache and the decrease in IL-6 in the blood were observed only in a subset of 86 patients infected during the second three waves of the pandemic. Kan Jang^®^ significantly increased physical activity and workout; however, it did not affect cognitive functions (attention and memory), quality of life score, inflammatory marker D-dimer, and C-reactive protein compared with the placebo group. ***Conclusions***. Overall, the results of this study suggest that Kan Jang^®^ is effective in treating mild and moderate COVID-19 irrespective of the SARS-CoV-2 variant of infection.

## 1. Introduction

The pathogenesis and progression of coronavirus infection is a multistep process [1,2], characterized by a wide variety of inflammatory symptoms and requires an appropriate therapeutic strategy, including targeting the initiation of defense response to the pathogen [3,4,5,6] and numerous extra- and intracellular interactions between components of host-defense and life-cycle regulation systems on all levels of regulation—genomic, transcriptomic, proteomic, metabolomic and macrobiotic [7]. Effective prevention or treatment of COVID-19 and other viral infections requires pharmaceutical intervention, which directly targets the virus and indirectly affects various defense systems mechanisms, including immune and inflammatory response, activation of detoxification processes, and reparation of oxidative stress-induced damages in compromised cells (Figure 1) [8]. This can be achieved with herbal preparations that have polyvalent and pleiotropic actions on host-defense systems by synergistically targeting multiple elements of molecular networks involved in inflammatory defense response; these are presumably more effective than mono-drugs that target only one receptor [9,10]. 

A blend of extracts of *Andrographis paniculata* (Burm. F.) Wall. Ex. Nees with *Eleutherococcus senticosus* (Rupr. & Maxim.) Maxim, known in Scandinavia under the proprietary name Kan Jang, was demonstrated to be effective in several clinical trials treating various viral upper-respiratory-tract infections (URTIs) [11,12,13,14,15,16,17,18]. It should be emphasized that Kan Jang and *A. paniculata* are two entirely different active substances of two different chemical compositions, with two unique pharmacological and toxicological profiles (biological signatures; Figure 2), which are different from their ingredients and purified compounds, e.g., andrographolides and eleutherosides [9]. The gene expression network and pharmacology network studies suggest a potential antiviral response, virus clearance, and antigen presentation due to the synergistic interaction of *A. paniculata* and *E. senticosus* (Figure 2). 

The evidence from many preclinical studies with *A. paniculata* and *E. senticosus* and their combination suggests that they can be used in prevention and treatment of viral infections, including coronavirus, at all stages of progression of inflammation, as well as in aiding the recovery of the organism by (i) modulating innate and adaptive immunity, (ii) anti-inflammatory activity, (iii) detoxification and repair of oxidative-stress-induced damage in compromised cells, (iv) direct antiviral effects of inhibiting viral docking or replication, and (v) improving quality of life during convalescence [8]. 

This study aimed to assess the clinical efficacy and safety of Kan Jang^®^ in 140 patients with mild or moderate COVID-19 inflammatory symptoms in the last three days who were not requiring intensive care unit (ICU) admission as per protocol.

Recently, we published the results of an interim analysis of this clinical trial in 86 patients recruited in Georgia during three waves of COVID-19 (Figure 3) [18]. 

“The latest wave caused several coronavirus variants of SARS-CoV-2 was characterized by a remarkable speed, lesser severity, comorbidity, lower death rate, and much higher positivity rate, but more cases with breathlessness, and characterized by newer symptoms like gastrointestinal, particularly in a younger population, as compared to the 1st wave” [19]. The further recruiting of patients in the ongoing study was conducted from 6 May 2022 due to the spread of the last wave of the SARS-CoV-2 variants pandemic in Georgia (Figure 3). 

This study provides the results of the analysis in the overall sample size of 140 patients included per protocol (IPP) and the intention to treat (ITT), including an additional subset of 54 patients admitted during the last wave of the pandemic, who were affected mainly by known and unknown mutants/variants of SARS-CoV-2.

## 2. Results

### 2.1. Patients

#### Demographic and Baseline Characteristics

One hundred and fifty-eight patients with confirmed diagnoses based on a positive SARS-CoV-2 test, experiencing mild to moderate COVID-19 symptoms [20], were screened for compliance with the inclusion criteria and assessed for eligibility. One hundred and forty patients with at least 3 to 8 symptoms (fatigue, headache, nasal discharge, loss of smell, taste, cough, muscle pain, and body temperature from 37 to 38 °C) for the last three days before admission to the hospital met the inclusion criteria and were randomly assigned to one of two treatment groups, Kan Jang^®^ (A) or placebo (B) (Figure 4). 

The groups did not show differences in baseline demographic, physical, or other critical clinical measurements, except for physical activity, muscle pain, loss of smell, and URTI scores, which were worse in the Kan Jang group than the placebo group (Table 1). 

Several blood parameters, specifically blood serum IL-6, D-dimer, C-reactive protein, neutrophils, lymphocytes, monocytes, eosinophils, and basophils counts, were substantially higher than the normal ranges typical for acute viral inflammation. However, no significant difference between groups for these inflammatory markers was observed (Table 1). 

### 2.2. Efficacy

The therapeutic efficacy of Kan Jang^®^ was assessed by comparing (i)—differences in the time to resolve inflammatory symptoms in Kan Jang^®^ and placebo groups of patients and (ii)—differences in the relief severity of inflammatory symptoms from the baseline in the Kan Jang^®^ and placebo groups of patients. Treatment groups were compared for all efficacy outcome measures to assess the primary and secondary endpoints.

#### 2.2.1. Primary Endpoints 

##### The Rate of Patients with Clinical Deterioration and Virus Clearance

In Kan Jang^®^ Group A of 68 patients, 12 were dropouts, and 3 were withdrawn from the study due to lack of efficacy and disease progression; they continued the treatment with steroids and antibiotics (Figure 5).

In Placebo Group B of 72 patients, 16 were dropouts, and 10 were withdrawn from the study due to lack of efficacy and disease progression; they continued the treatment with steroids and antibiotics (Figure 5a). 

The disease progression rate in the placebo group (56 patients) was 17.86%, while in the Kan Jang group (56 patients) it was 5.36% (*p* = 0.0176, significant result at 95% significance level (confidence 95%), power—92.99%, IPP analysis; Figure 5a). The statistical difference (st. error 0.0477) in the rates of patients with clinical deterioration in the Kan Jang treatment (A) and placebo control (B) groups was significant (*p* = 0.0236 in 140 patients included in ITT analysis, observed power—90.66%).

The overall clinical effectiveness, defined as the ratio of proportions of effective to ineffective cases in the Kan Jang (94.54/5.36 = 17.65) vs. placebo control (82.14/17.86 = 4.6) groups, was 3.84. Absolute risk reduction (ARR%) by Kan Jang treatment was 12.5%, while the relative risk reduction was 214.8%. The number of patients that we needed to treat with Kan Jang to prevent one additional bad outcome (defined as the number we needed to treat, NNT = 1/ARR) was eight.

The percentage of patients with a negative SARS-CoV-2 virus test was 17% lower in the Kan Jang group compared to the placebo group after 14 days of the treatment (19.3% vs. 36.3%, difference—17.0 ± 8.5%, *p* = 0.819; Figure 5b). The rate of virus clearance was faster at 1.6 days shorter in the Kan Jang group (10.8 days) compared to the placebo group (12.4 days), along with the time to virus clearance in 50% of patients (hazard ratio Kan Jang/placebo = 1.686, 95% CI of ratio from 0.8698 to 3.269; Figure 5b). 

##### Recovery Time and Time to Fever Resolution

The proportion of patients with high body temperature (from >37 °C to <38 °C) was 17.3% lower in the Kan Jang group (29.5%) compared to the placebo group (46.8%) after one week of treatment; however, this difference was statistically insignificant (Figure 6a). The duration of increased body temperature (from >37 °C to <38 °C) was also shorter in the Kan Jang group compared to the placebo group; median recovery in the Kan Jang^®^ group was six days while in the placebo group it was 7 days (hazard ratio Kan Jang/placebo = 1.336, 95% CI of ratio from 0.808 to 2.309; Figure 6a).

The median (50%) number of patients who elapsed at the clinic was reached at three days less for the Kan Jang treatment (11 days) compared to the placebo (14 days) (hazard ratio Kan Jang/placebo = 0.7857, 95% CI of ratio from 0.530 to 1.164; hazard ratio Kan Jang/placebo = 1.232, 95% CI of ratio from 0.778 to 1.951; Figure 6b). 

##### The Severity and Time to Resolution of Inflammatory Symptoms

The occurrence of various inflammatory symptoms differed at the baseline in the cohort of COVID-19 patients recruited in this study (Table 1); the most common symptoms were fatigue (in 100% of patients), headache (in 91% of patients), sore throat (in 62% of patients), cough (in 55% of patients), and muscle pain (51%), while other symptoms were less frequently observed—rhinorrhea (27%) and loss of smell (16%) and taste (4%). 

The results of the assessment of upper respiratory symptoms using the Wisconsin URTI survey (including runny and plugged nose, sneezing, sore throat, cough, hoarseness, headache and congestion, and feeling tired) show a beneficial effect of Kan Jang compared to the placebo (Figure 7).

The severity of all inflammatory symptoms gradually decreased from the baseline to the end of therapy (Day 14) and the follow-up period (21 days) in both groups of patients (Figure 7, Figure 8, Figure 9, Figure 10 and Figure 11). However, the relief from inflammatory symptoms for patients in the Kan Jang group was significantly better compared to placebo effects over the time of the treatment and follow-up, including relief of sore throat (Figure 8), rhinorrhea (nasal discharge/runny nose, Figure 9), myalgia (muscle pain, Figure 10), and cough (Figure 11).

The rates of resolution of cough over time were similar in the Kan Jang and placebo groups (Figure 11a); however, the relief from severity of the cough for patients taking Kan Jang over the time from the baseline was significantly more effective compared to the placebo (Figure 11b). Kan Jang^®^ relieved the severity of cough compared to the placebo, as was evident in the ITT analysis in 140 randomized patients (*p* = 0.053), in the IPP analysis in 100 patients who completed the study (*p* = 0.018), in a subset of 54 patients during last wave of the COVID-19 pandemic in Georgia (*p* = 0.0049, ITT analysis), and in a subset of 42 patients who completed the study during the last wave of the COVID-19 pandemic in Georgia (*p* = 0.0047, IPP analysis). 

#### 2.2.2. Secondary Endpoints

Secondary endpoints comprised the measures of (i)—immune response marker IL-6 concentration in the serum, (ii)—blood hypercoagulation marker D-dimer, (iii)—inflammatory marker C-reactive protein, (iv)—physical activity, (v)—physical performance, and (vi)—cognitive performance.

##### Blood Serum Markers of Immune Response and Inflammation

At the beginning of the study, the baseline levels of all selected markers of immune response and inflammation—IL-6, C-reactive protein, and D-dimer—were considerably higher than typical blood values (Table 1). In 3 days of the treatment with Kan Jang, the level of IL-6 reached the standard limit of 7 pg/mL, while in the placebo group, 14 days after randomization (Figure 12a). 

Between-groups comparison of the changes in the level of cytokine IL-6 in the blood did not show significant with treatment time (*p* = 0.1619), despite the Kan Jang^®^ treatment showing a noteworthy decrease in cytokine IL-6 in the blood (−11.575 pg/mL on day 14) compared to the placebo (−2.246 pg/mL on day 14; Figure 12b, ITT analysis).

The levels of C-reactive protein and D-dimer were also normalized in the blood of all patients at the end of treatment; however, the Kan Jang^®^ treatment made no significant difference in the effects on C-reactive protein and D-dimer compared to the placebo (*p* > 0.05, Figures in Appendix A).

##### Physical Activity and Physical and Cognitive Performance

The Kan Jang^®^ treatment significantly increased physical activity (*p* = 0.0023) and workout time (in min, *p* = 0.020) in patients compared to the placebo (Figure 13).

No significant differences between the effects of Kan Jang^®^ and placebo were observed on cognitive performance, quality of life, and other secondary outcomes.

### 2.3. Safety

The extent of exposure to Kan Jang was 84 capsules consumed in 14 consecutive days, in a total of 1.26 g andrographolides. The dose was 6 capsules daily in a total of 1672.5 mg of the fixed combination of 1560 mg Herba Andrographis standardized native extract (corresponding to 90 mg of andrographolides) and 112.5 mg Radix Eleutherococci native extract, which is equivalent to 7–12.6 g and 1.9–3.4 g of herbal substances, correspondingly. 

Regardless of causality, adverse events were monitored for all patients from the first dose and through the one-week follow-up period. No adverse events of allergic reactions like urticaria, angioedema, paresthesia, anaphylactic reactions, rush, and pruritus (itchy skin) were observed after initiation of study treatments, including events likely to be related to the underlying disease or likely to represent concomitant illness (Appendix A). Deaths, other serious adverse events, and other significant adverse events that deserve special attention were not recorded in the 140 patients during the treatment in either study group. The results of blood analysis showed the normalizing anti-inflammatory effect of Kan Jang. Vital signs related to safety did not reveal evidence of a drug effect. 

## 3. Discussion

Several antiviral drugs (Paxlovid™ [21], Remdesivir, Bebtelovimab [22], Molnupiravir [23], Andrographis [24,25,26]) and immune modulators (Olumiant, an interleukin-6 receptor blocker [27]) were approved by drug regulatory authorities for use against COVID-19 for treating non-hospitalized or hospitalized COVID-19 patients. Recently, a systematic review and meta-analysis of 50 clinical trials (involving 11,624 patients) of Chinese herbal medicine in COVID-19 concluded that they are effective and safe in combination with conventional “Western” drugs, but the certainty of the evidence ranged from moderate to very low [23]. Similar conclusions were made in other systematic reviews and meta-analyses of Chinese herbal medicine in COVID-19 [28,29,30,31,32].

Herba Andrographidis preparations are the first herbal medicine for treating COVID-19 and were formally adopted in Thailand [24,25,26] despite contradictory conclusions from the efficacy and safety results in clinical studies [33,34,35,36,37,38,39,40]. 

In this clinical trial, the efficacy of Kan Jang in patients with mild or moderate symptoms of COVID-19 was studied. In the first subset of 86 patients, Kan Jang was found to effectively relieve sore throat, muscle pain, and runny nose in patients with mild or moderate symptoms of COVID-19 [18]. Further increase in the sample size to 140 patients improved Kan Jang’s efficacy in ameliorating cough, sore throat/pain, runny nose, and muscle soreness and reducing the Wisconsin Upper Respiratory Symptoms score compared to the placebo. Kan Jang^®^ decreased blood inflammatory marker IL-6 over the recovery time in the subgroup of 86 patients with mild or moderate symptoms; however, the effect was statistically insignificant in the sample size of 140 patients, which was enriched with the 54 patients infected with a less pathogenic variant during the last wave of COVID-19 (Figure 3).

Kan Jang^®^ increased physical activity and workout; however, it did not affect cognitive functions (attention and memory), quality of life score, time to normalization of body temperature, inflammatory marker D-dimer, or C-reactive protein compared to the placebo group.

Notably, the rate of patients with clinical deterioration and disease progression was significantly lower in the Kan Jang group than in the placebo group. The disease progression rate in the placebo group was about 2.5-fold higher than in the Kan Jang Group (Figure 5). The absolute risk reduction (ARR) via Kan Jang treatment was 14%, and the relative risk reduction (RRR) was 243.9%. The number of patients we needed to treat (NNT) with Kan Jang to prevent one additional bad outcome was found to be eight patients. As a rule, the higher the ARR and lower the NNT, the more effective the intervention. For comparison, Paxlovid™ effectively reduced the risk of progression (ARR of 6.2%) to severe COVID-19 in symptomatic adults at high risk of progression, and the number we needed to treat (NNT) was 17 patients [21]. Paxlovid™, unlike Kan Jang, induces dysgeusia, diarrhea, hypertension, and myalgia [21]. The safety of Kan Jang is an essential advantage as it is greater than the safety of other anti-COVID-19 drugs [21,41,42]. 

The treatment with Kan Jang^®^ was well tolerated, and no adverse events were recorded in this study. That is consistent with the results of previous clinical studies where only four adverse events were recorded in one [17] of six studies [11,12,13,14,15,16,17] with 494 patients in the Kan Jang group. In that study, the only common adverse reaction was mild pruritus observed in four patients in the Kan Jang^®^ group and six in the placebo group [17], and no severe adverse reactions were observed. There have been 0.706% adverse events in 562 patients in all seven clinical studies with Kan Jang^®^ groups, including our recent study. That is significantly less than 131 mild to moderate adverse events observed in 690 (19%) patients from ten studies on *A. paniculata* herbal preparations that are essentially safe [43], and 383 (4.04%) adverse events in 9490 participants using andrographolide derivative injections, which can be life-threatening, mainly due gastrointestinal, skin and subcutaneous tissue disorders, as well as anaphylaxis [43].

Kan Jang has an excellent safety profile [16,17], presumably due to the contribution of the adaptogenic and antitoxic activity of Eleutherococcus [44]. Cytoprotective (neuroprotective, hepatoprotective, cardioprotective), stress-protective, antioxidant, antitoxic, and immunomodulating activity of Eleutherococcus preparations was demonstrated in many experimental models on isolated cells and (in vitro and ex vivo) animals [45,46,47,48]. This evidence of the safety of the Kan Jang combination shows that the pharmacological activity and toxicity of multi-component drugs are different from their ingredients, and the effects observed on purified compounds cannot be just extrapolated to their combinations with other substances, or vice versa, due to their multiple synergistic and antagonistic interactions in the organism [9,46,47,48] (Figure 2). In other words, the results obtained in this study are product-specific. They cannot be applied to mono herbal drugs, e.g., Andrographis extract, which in turn is the mixture of 39 compounds such as andrographolides, flavonoids, etc., and deregulates quite a different number of genes than expected from a simple calculation of many constituents of the plant extracts [9]. 

In this context, an essential difference in the modes and mechanisms of action of Kan-Jang and Andrographis compared with other drugs is their multitarget effects both directly on virus-receptor binding, viral membrane fusion into the host cell, viral replication, transcription, translocation, assembly, and release to infect other host cells [8,49], as well as indirect antiviral activity due to multiple effects on innate and adaptive immune systems, inflammatory response, and essentially on recovery of oxidative-stress-induced damage in compromised cells [8] (Figure 1).

The results of this study are consistent with the previous publications [11,12,13,14,15,16,17,18], where Kan Jang effectively relieved a sore throat, runny nose, cough, and muscle pain in patients with various upper-respiratory-tract viral infections. Kan Jang effectively ameliorated these URTI symptoms irrespective of the virulence and potential morbidity of different strains/variants of the SARS-CoV-2 virus, which were inducing mild to moderate COVID-19.

The limitations of our study were the short duration of symptoms of COVID-19 before treatment (three days) and the lack of concomitant chronic diseases, which can increase the risk of progression of the disease and intensive care therapy for patients. Further studies are required in patients at high risk of the progression of pneumonia, acute respiratory distress syndrome, and septic shock.

## 4. Materials and Methods

The methods of the study were described according to CONSORT recommendations [50] in a recent publication in detail, including study design, recruitment, and screening of patients, schedule of examinations, study population, patient inclusion and exclusion criteria, participant withdrawal, the intervention and comparator, doses and treatment regimens, methods of randomization and blinding, generation of sequence, allocation concealment, implementation, blinding of participants and personnel, completeness of outcome data, evaluation of compliance, statistical analysis, and sample size considerations [18] (Appendix A). 

### 4.1. Study Design, Recruitment, Screening of Patients, and Schedule of Examinations

This prospective, randomized, placebo-controlled, quadruple-blind, two-parallel-group (Figure 4 and Appendix A), phase II interventional study was conducted at Tbilisi State Medical University, Tbilisi, Georgia, with the approval of the Biomedical Research Ethics Committee of Tbilisi State Medical University and the National Council on Bioethics (registration no. 3-2021/87, date of final protocol approval 25 March 2021). The ClinicalTrials.gov identifier is NCT04847518 (https://www.clinicaltrials.gov/ct2/show/NCT04847518; accessed on 6 June 2022). 

Recruitment for the study was initiated on 26 May 2021, the 86th patient was recruited on 30 March 2022, and the study was completed on 30 October 2022. 

Overall, 158 patients were assessed for eligibility; 140 patients met the inclusion criteria of mild COVID-19 symptoms [20] and were randomized and included in the ITT analysis of the study (Appendix A, Full Analysis Dataset). In total, 100 patients (71.4% of randomized) completed their respective treatment cycles according to protocol, while 40 patients (38.6% of randomized) discontinued therapy after receiving at least one dose of study preparations at the patient’s request. Nighty-nine patients, who completed the treatment, were evaluated for treatment efficacy for two weeks (visits 2 and 3) of treatment and one week after completing the treatment (the follow-up visit 4). The distribution between the study groups and the disposition of patients are shown in the flow chart in Figure 4. 

### 4.2. Datasets Analyzed and Evaluation of Compliance 

One hundred patients enrolled in the screening were assigned to randomized investigational products and included in the ITT analysis to ensure they met the inclusion criteria and to avoid potential bias due to the exclusion of patients at the baseline (visit 1). The compliance was assessed by a count of the returned capsules, regardless of when the dropout patients terminated the treatment. Overall compliance in each group was more than 80%, comprising 89.0 ± 2.8% in the Kan Jang group (*n* = 68) and 82.0 ± 3.4% in the placebo group (*n* = 72). There was an insignificant difference between groups (*p* = 0.2887) in the ITT analysis (Appendix A). 

An efficacy subset analysis per protocol (PP) was performed in patients when they completed the study therapy (visits 3 and 4). A per protocol (PP) analysis aims to identify a treatment effect on the symptoms. Therefore, some patients were excluded from the complete analysis set (ITT), and the PP population was used for the PP analysis, ensuring 100% compliance (Figure 4). The schedule of procedures and examinations is shown in Table 2.

The medication compliance was calculated by counting the remaining capsules for each subject from the first to the last day of the study and verified by the study monitor at the end of the treatment. All the patients were provided with diary cards on which the daily usage of study medication was recorded. The duration of the treatment was recorded. Unused capsules were returned to the sponsor.

### 4.3. Study Population and Inclusion and Exclusion Criteria

The population for this study (aged 18 years and older of either sex; mean age: 47.98 ± 17.40 years) consisted of COVID-19 patients in a stable, moderate condition (i.e., not requiring intensive care unit (ICU) admission) with confirmed diagnoses based on positive SARS-CoV-2 tests and at least three of eight mild to moderate COVID-19 symptoms [20]: fatigue, headache, sore throat, nasal discharge, cough, pain in muscles, loss of taste and loss of smell (Table 1) in the last three days before recruitment for the study. Subjects were required to be under observation in the hospital, be able to take medication alone and give written informed consent. 

Patients admitted with severe acute respiratory syndrome under invasive mechanical ventilation and diagnosed with an etiologic agent other than COVID-19 were excluded from the study. Other exclusion criteria were: acute and chronic pulmonary diseases, chronic rhinosinusitis, renal failure or creatinine 2.0 mg/dL, type 2 diabetes, autoimmune disease, patients taking antibiotics for a reason other than COVID-19 at enrollment, chronically suppressed immune system (AIDS, lymphoma, corticosteroid therapy, chemo- or radiotherapy for last six months), cancer patients taking immunosuppressive drugs, pregnant or lactating women, and subjects participating in other clinical studies or taking any medication influencing the outcome measures during the clinical trial. 

Patients were free to withdraw from the clinical study at any time without giving a reason.

### 4.4. Intervention, Comparator, Doses, and Treatment Regimens

Pharmaceutical-grade standardized extracts of *A. paniculata* L. Nees. (herb) and *E. senticosus* (root) genuine extracts, as well as their fixed combination, Kan Jang^®^, were manufactured, tested, and then released for human use as per the ICH Q7A and EMEA guidelines for Good Agricultural and Collecting Practice and Good Manufacturing Practice (GMP) of active pharmaceutical ingredients at the Swedish Herbal Institute, which holds a valid EU-GMP license to produce pharmaceuticals.

One capsule of Kan Jang^®^ (size 0, batch no. 40154, expiry date: February 2024) contains 260 mg of *A. paniculata* native extract SHA-10 (drug–native extract ratio of 4.5–8.1:1, extraction solvent 70% ethanol), including 15 mg of diterpene lactones (andrographolide and 14-deoxy-11,12-didehydroandrograholide), and 18.75 mg of *E. senticosus* native extract (drug–native extract ratio of 17–30:1, extraction solvent 70% ethanol, 0.25 mg Eleutherosides B and E); for details, see Appendix A in previous publication [18]. The matrix contains inactive excipients (microcrystalline cellulose and magnesium stearate). The placebo capsules (size 0, batch no. 40154) containing the inactive excipients were identical to the Kan Jang^®^ capsules. Both preparations’ appearance, smell, and color were similar and organoleptically undistinguishable.

The daily dose of the study intervention was two capsules three times per day for two consecutive weeks with a daily intake of 1560 mg of the dry extract *A. paniculata* SHA-10, corresponding to 90 mg of diterpene lactones andrographolides from *A. paniculata* herb and 112.5 mg of dry extract of E. senticosus, in the Kan Jang^®^ group. All the patients were provided with diary cards on which the daily consumption of the study preparation was recorded. The number of capsules consumed during the treatment by each patient was verified for compliance according to the duration of the treatment/illness (days at the clinic). The Principal Investigator was responsible for maintaining drug accountability records.

### 4.5. Randomization, Blinding, and Allocation Concealment

Study preparations were randomly labeled by a qualified pharmacist (QP), and the random sequence of treatment codes was retained at the manufacturing site until the study was completed. Randomization, blinding, and allocation concealment were performed as previously reported [18], ensuring a quadruple-blind design.

#### 4.5.1. Randomization and Blinding

Study preparations were labeled by a quality-assurance-responsible person (QP) at the manufacturing site. The QP used a random numbers sequence, which was generated by the PRISM GraphPad software (2017 Online version, GraphPad Software Inc., San Diego, CA, USA) “random number generator” (https://www.graphpad.com/quickcalcs/randomize1.cfm, accessed on 16 February 2022). The randomization sequence comprises a table of two columns (A and B) filled with randomly distributed unique numbers from 1 to 160.

It provides information on the content of each package—how placebo and verum packages/containers were labeled. The QP assigned the codes A and B to Kan Jang and placebo sets of packages.

#### 4.5.2. Allocation Concealment

All the packages had allocation concealed random numbers printed on the label.

The PI blinded the randomization sequence list and code, so the PI assigned the packages sequentially as per numbers. The random sequence was disclosed to the statistician before the statistical evaluation of the results when 86 patients had completed the treatment.

#### 4.5.3. Implementation and Blinding

At the first visit, participants received a consecutive number starting from 1 to 86. They were linked to a unique number according to the randomization sequence. Patients were sequentially enrolled by the PI, assigned a random number, and received the capsules in the package. The Investigator produced the participant list and gave a treatment code number (from 1 to 140) to each patient. They recorded the patients’ names on case report forms (CRFs) and on the labels of the packages. The table shows the names of patients and corresponding study preparation numbers (treatment numbers mentioned on the labels of boxes).

Blinding for trial subjects was achieved using labeled packages containing capsules of the same appearance. The study product was delivered to the clinic prelabeled and coded according to the randomization list. The Principal Investigator kept the study participants’ list, identifying the patients and the investigational product packages (numbers). The treatment code providing information about the actual assignment of groups A and B to Kan Jang^®^ and placebo was broken by the QP. After that, a statistical analysis of the datasets was completed, and the results of the study were obtained. Thus, the study was quadruple-blind since study preparations were blinded to all investigators, care providers, participants, and outcome assessors.

### 4.6. Efficacy and Safety Outcomes and Endpoints

The efficacy endpoints were the differences in duration and the relief of inflammatory symptoms in Kang Jang and placebo group patients. The outcomes included changes in the severity of inflammatory symptoms, measured using various scales’ scores, from the baseline to the end of therapy (Day 14) and follow-up period (21 days after randomization). 

The primary efficacy outcome measures of the study were: (i)—the rate of patients (%) with clinical deterioration, (ii)—duration of hospitalization, (iii)—the time to virus clearance, (iv)—the duration of the acute phase of disease assessed as the time from the start of study medicine to complete symptom resolution, (v)—fever resolution and relief from the severity of fatigue, headache, sore throat, cough, rhinorrhea (nasal discharge/runny nose), myalgia (muscle pain), and loss of smell and taste.

Secondary endpoints comprised the measures of (i)—immune response marker IL-6 concentration in the serum, (ii)—blood hypercoagulation marker D-dimer, (iii)—inflammatory marker C-reactive protein, (iv)—physical activity, (v)—physical performance, (vi)—cognitive performance, and (vii)—severity of respiratory symptoms and quality of life according to the Wisconsin Upper Respiratory Symptoms score. 

Safety and tolerability were assessed by monitoring the incidence and duration of adverse events. Laboratory samples for safety lever enzyme tests of aspartate aminotransferase (AST) and alanine aminotransferase (ALT) were taken on the day of inclusion in the study. All safety samples were drawn using standard syringes and analyzed according to the regular laboratory routine. Data were printed on separate sheets, one sheet per patient per test day. The data were published [18] in the CRF (see Appendix A in previous publication [18]). Subjects were asked to report any adverse events they experienced to the Principal Investigator. For each patient, code envelopes were made. They contained information about the treatment the patient received during the study. The envelope was to be opened only in case of emergency, such as if severe side effects occurred. Adverse event reporting was conducted on visits 2, 3, and 4. The subjects were told to report any AE occurring during the study to the Investigator or the Investigator’s personnel. Open, standardized AE questioning such as “Have you had any health problems since you were last questioned?” was performed by the Investigator (or designee) at each contact with the subject. That open, standardized questioning was carried out discreetly to prevent the patients from influencing each other.

## 5. Conclusions

This study provides new evidence on the clinical efficacy and safety of adaptogens, specifically Kan Jang^®^, in the acute phase of mild COVID-19. Kan Jang^®^ reduces the risk of disease progression, the duration of illness, virus clearance, and days of hospitalization, and accelerates recovery of patients, relief of sore throat, muscle pain, and runny nose, and normalization of body temperature. Kan Jang^®^ significantly relieves the severity of inflammatory URTI symptoms, including sore throat, cough, runny nose, and muscle pain, and increases patients’ physical performance (workout) compared to the placebo. 

## Figures and Tables

**Figure 1 pharmaceuticals-16-01196-f001:**
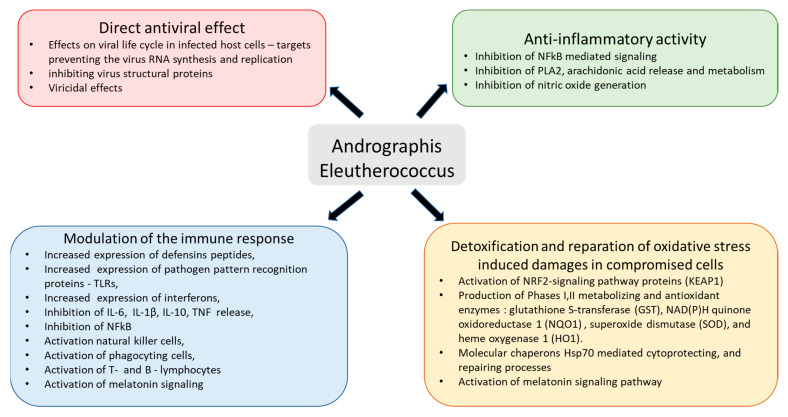
Schematic diagram of reported effects of Andrographis and Eleutherococcus elucidated in animal and cell culture models: (i) modulatory effects on immune response (blue block), (ii) anti-inflammatory activity (green block), (iii) detoxification and repair of oxidative-stress-induced damage in compromised cells (brown block), and (iv) direct antiviral effect via infraction with viral docking or replication (red block), modified from Panossian and Brendler [8].

**Figure 2 pharmaceuticals-16-01196-f002:**
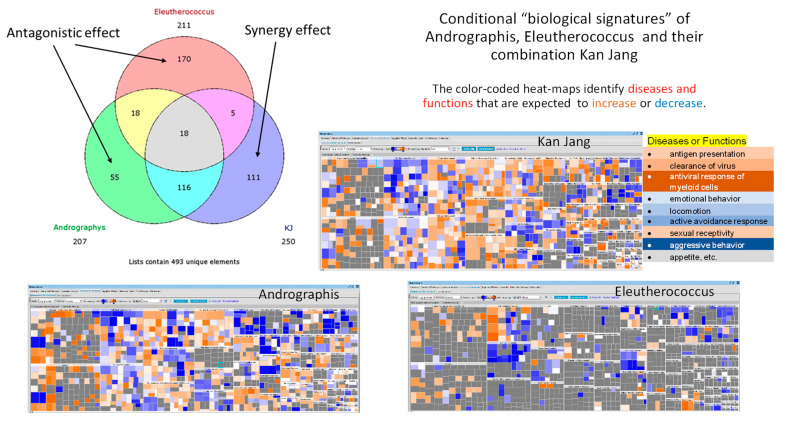
Venn diagrams show the number of deregulated genes in neuroglia cells in response to Andrographis, Eleutherococcus, and their fixed combination Kan Jang (KJ). The diagram includes overlapping sections and sections associated with synergistic interactions of Andrographis and Eleutherococcus in Kan Jang (111 unique genes) [12]. Conditional “biological signatures” are shown in the form of color-coded heatmaps, which identify the diseases and biological functions associated with gene expression in response to Andrographis, Eleutherococcus, and Kan Jang.

**Figure 3 pharmaceuticals-16-01196-f003:**
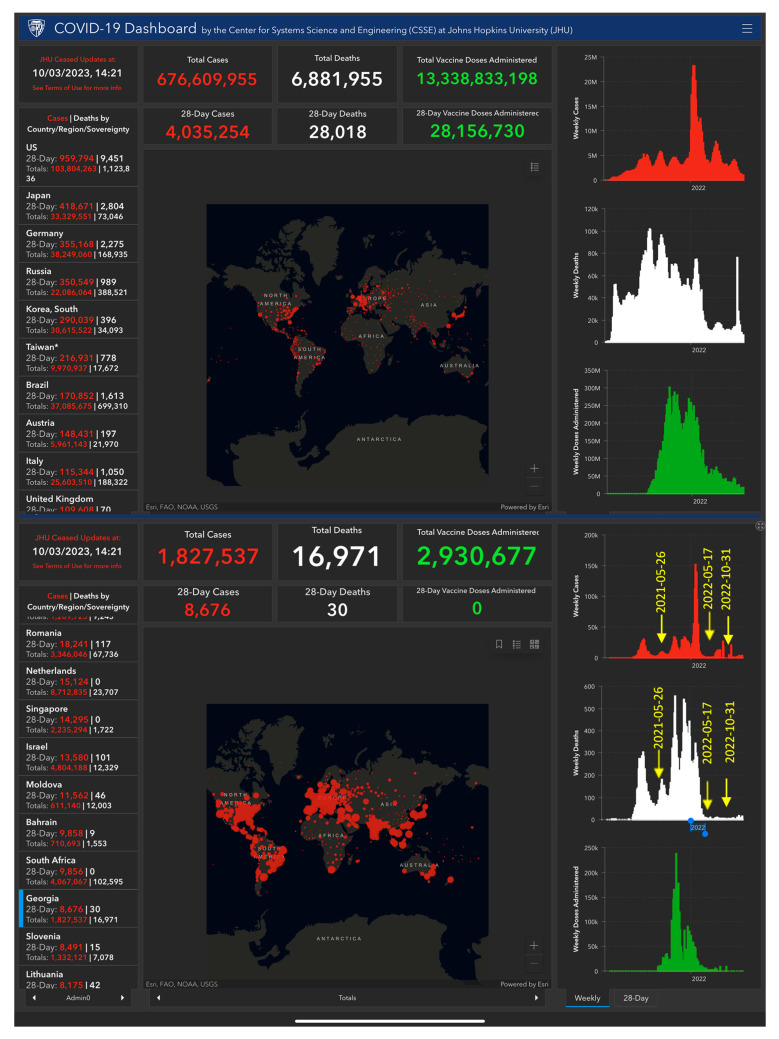
The figure shows six waves of the COVID-19 pandemic, including the number of infected patients recorded over time (in red color charts) in Georgia and worldwide. In white colors are the number of deaths, and the number of vaccinations are in green colors. Eighty-six patients were studied during three waves characterized as the most severe period of the pandemic from 26 May 2021 to 30 March 2022, while a subset of 54 patients was studied on the last wave of less harmful SARS-CoV-2 variants’ infection from 17 May 2022 to 30 October 2022. All 140 patients were diagnosed as having mild or moderate COVID-19 according to the WHO classification [16]. The background of Figure 3 was made using a screenshot of the COVID-19 Dashboard from the Center for Systems Science and Engineering (CSSE) at Johns Hopkins University (JHU). This is available online from https://www.arcgis.com/apps/dashboards/bda7594740fd40299423467b48e9ecf6; (accessed on 28 July 2023), and the details can be seen in high-resolution and real-time online.

**Figure 4 pharmaceuticals-16-01196-f004:**
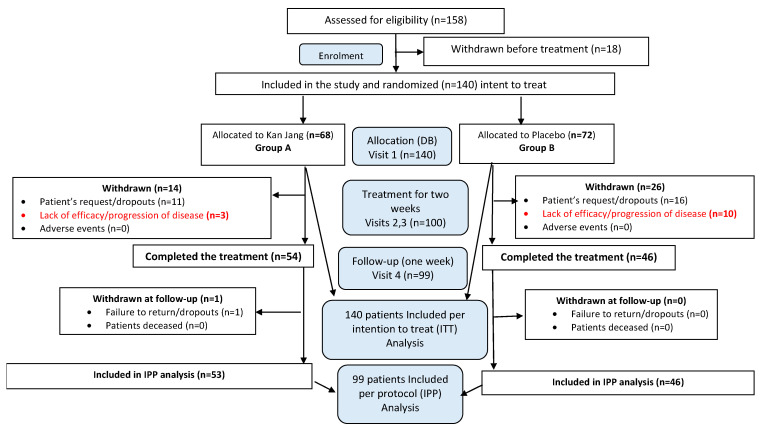
Schematic diagram of the trial. For details of the disposition of patients, see Appendix A.

**Figure 5 pharmaceuticals-16-01196-f005:**
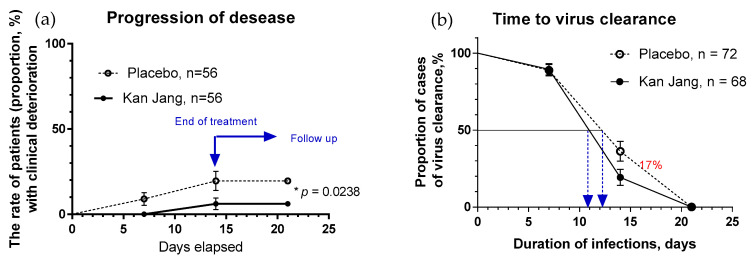
(**a**)—The rates of patients with clinical deterioration in the treatment and control groups; hazard ratio Kan Jang/placebo = 0.2906, 95% CI of ratio from 0.094 to 0.894. (**b**)—The virus clearance in the treatment and control groups: Kaplan–Meier curves show the percent of patients with SARS-CoV-2 virus over the time from randomization (Day 1) to the end of the treatment (Day 14) and the follow-up period for one week (Day 21) in the treatment and control groups; hazard ratio Kan Jang/placebo = 1.686, 95% CI of ratio from 0.8698 to 3.269. * *p* < 0.05.

**Figure 6 pharmaceuticals-16-01196-f006:**
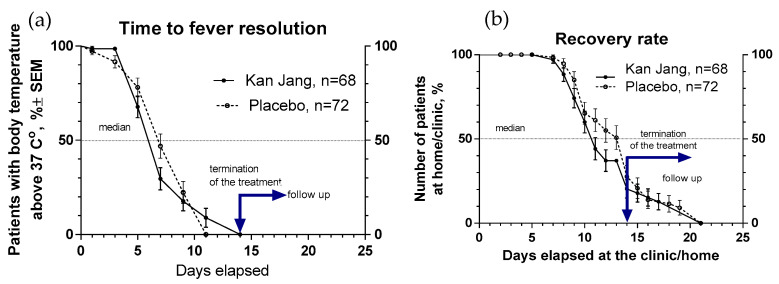
(**a**)—Duration of increased body temperature (from >37 °C to <38 °C) in the treatment and control groups; median recovery: Kan Jang^®^—6 days, placebo—7 days; hazard ratio Kan Jang/placebo = 1.336, 95% CI of ratio from 0.808 to 2.309. (**b**)—Duration of hospitalization in the treatment group and control group; Kaplan–Meier curves show the percent of patients hospitalized over the time from randomization (Day 1) to the end of the treatment (Day 14) and followed up for one week (Day 21) in the treatment and control groups.

**Figure 7 pharmaceuticals-16-01196-f007:**
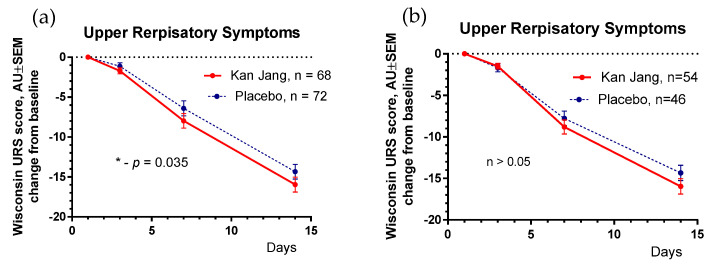
(**a**)—Between-groups comparison of the changes from the baseline of Wisconsin Upper Respiratory Symptoms scores in patients of group A (Kan Jang) and group B (placebo) over the time from Day 1 to Day 21 shows the statistically significant positive effect of Kan Jang treatment vs. placebo both (**a**)—in 140 patients included in ITT analysis (*p* = 0.035), and (**b**)—in 100 patients included in IPP analysis (*p* > 0.05); * *p* < 0.05.

**Figure 8 pharmaceuticals-16-01196-f008:**
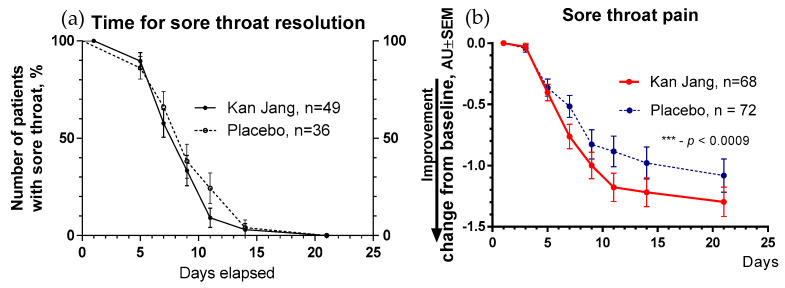
(**a**)—Time to relieve sore throat in the treatment and control groups: Kaplan–Meier curves show the percent of patients with a sore throat over the time from randomization (Day 1) to the end of the treatment (Day 14) and follow-up for one week (Day 21); median recovery, Kan Jang^®^, —7 days, placebo—11 days; hazard ratio Kan Jang/placebo = 1.36, 95% CI of ratio from 0.7350 to 2.516. (**b**)—Relief of the sore throat; the changes in the severity of the symptom from the baseline of patients in group A (Kan Jang) and group B (placebo) over the time from Day 1 to Day 21. Between-groups comparison of the changes in the symptom severity from the baseline over time shows significant interaction (*p* = 0.0009). The Kan Jang^®^ treatment has a statistically significant effect on the relief of the sore throat compared to the placebo in the ITT analysis. *** *p* < 0.0001.

**Figure 9 pharmaceuticals-16-01196-f009:**
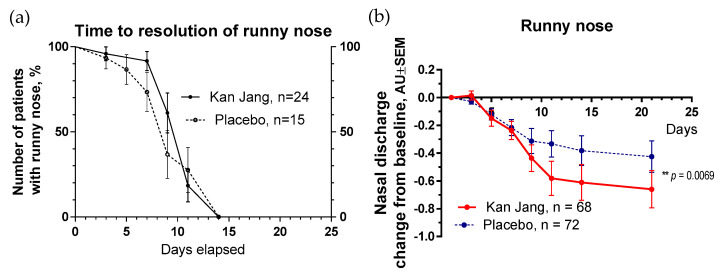
(**a**)—Time to a resolution of runny nose in the treatment and control groups: Kaplan–Meier curves show the percent of patients with a runny nose over the time from randomization (Day 1) to the end of the treatment (Day 14) and follow-up for one week (Day 21) and in the treatment and control groups; median recovery: Kan Jang^®^—14 days, placebo—14 days; hazard ratio Kan Jang/placebo = 0.672, 95% CI of ratio from 0.2321 to 1.944. (**b**)—Reduction in nasal discharge; the changes in the severity of the symptom from the baseline of patients in group A (Kan Jang) and group B (placebo) over the time from Day 1 to Day 21. Between-groups comparison of the changes in the symptom severity from the baseline over time shows significant interaction *(p* = 0.0069). The Kan Jang^®^ treatment has a statistically significant effect on the reduction in nasal discharge compared to the placebo ITT analysis. ** *p* < 0.01.

**Figure 10 pharmaceuticals-16-01196-f010:**
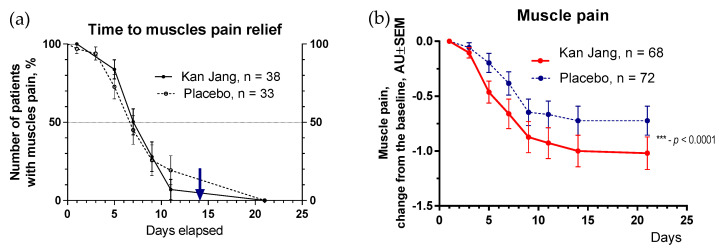
(**a**)—Time to muscle pain relief in the treatment and control groups. Kaplan–Meier curves show the percent of patients with muscle pain over the time from randomization (Day 1) to the end of the treatment (Day 14) and follow-up for one week (Day 21); median recovery, Kan Jang^®^—9 days, placebo—11 days; hazard ratio Kan Jang/placebo = 0.8965, 95% CI of ratio from 0.4434 to 1.812. (**b**)—Relief of the muscle pain; the changes in the severity of the symptom from the baseline of patients in group A (Kan Jang) and group B (placebo) over the time from Day 1 to Day 21. Between-groups comparison of the changes in the severity of the symptom from the baseline over time in 140 patients included in the ITT analysis shows significant interaction (*p* < 0.0001). Statistical significance of the effects of Kan Jang^®^ treatment vs. placebo on muscle pain relief is shown as: *** *p* < 0.0001.

**Figure 11 pharmaceuticals-16-01196-f011:**
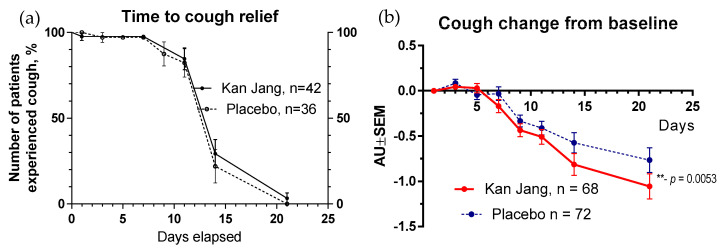
(**a**)—Time to resolution of cough in the treatment and control groups: Kaplan–Meier curves show the percent of patients with a cough over the time from randomization (Day 1) to the end of the treatment (Day 14) and follow-up for one week (Day 21) and in the treatment and control groups; hazard ratio Kan Jang/placebo = 0.7027, 95% CI of ratio from 0.2744 to 1.799. (**b**)—Relief of the cough; the changes in the severity of the symptom from the baseline of patients in group A (Kan Jang) and group B (placebo) over the time from Day 1 to Day 21. Between-groups comparison of the changes in the symptom severity from the baseline over time shows significant interaction (*p* = 0.0053). The Kan Jang^®^ treatment significantly relieved the cough compared to the placebo in the ITT analysis, ** *p* < 0.01.

**Figure 12 pharmaceuticals-16-01196-f012:**
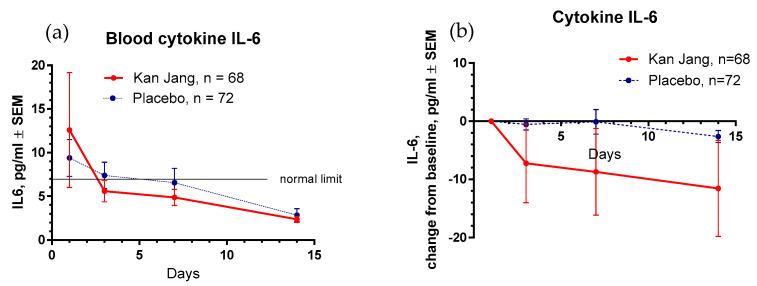
(**a**)—Concentration of IL-6 (mean ± SD) in the blood of patients in group A (Kan Jang) and group B (placebo) over the time from Day 1 to Day 14. (**b**)—The changes from the baseline in the levels (mean ± SD) of cytokine IL-6 in the blood of patients in group A (Kan Jang) and group B (placebo) over the time from Day 1 to Day 14. Between-groups comparison of the changes in the level of cytokine IL-6 in the blood from the baseline over time shows an insignificant difference (*p* = 0.1619) between groups A and B.

**Figure 13 pharmaceuticals-16-01196-f013:**
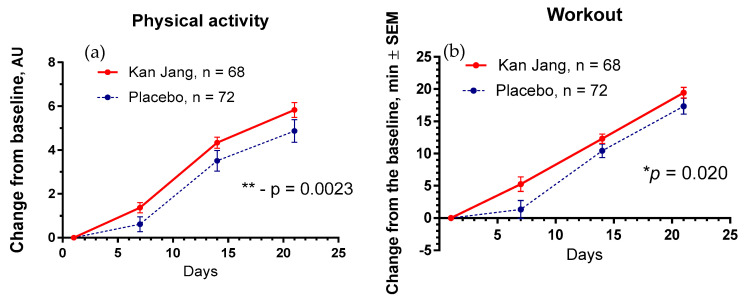
Between-groups comparison of the changes from the baseline in (**a**)—the overall physical activity and (**b**)—physical performance/workout time (in min) of patients in group A (Kan Jang) and group B (placebo) over the time from Day 1 to Day 21 shows the statistically significant positive effect of Kan Jang treatment vs. placebo in the 140 patients included in the ITT analysis. * *p* < 0.05, ** *p* < 0.001.

**Table 1 pharmaceuticals-16-01196-t001:** Baseline demographic characteristics, outcome measures, and laboratory biochemical and hematological measurements.

		Unit		Group A Kan Jang			Group B Placebo		Signif. of Difference
Parameters			n	Mean	SD.	n	Mean	SD.	*p*-Value
Age		Years	68	50.35	17.82	72	45.74	16.82	0.143 ^b^
Gender, Male/Female, %—53/87 = 61%		Male/Female	68	28/40 = 70%		72		25/47 = 53%	0.785
BMI		kg/m^2^	68	24.58	3.274	72	24.52	3.376	0.908 ^b^
Start of symptoms		days	68	<3		72	<3		
Viral load, SARS-CoV-2		%	68	100		72	100		
Body temperature		°C	68	37.86	0.58	72	37.7	0.59	0.148 ^b^
Fatigue	100% patients	A.U.	68	1.794	0.407	72	1.819	0.539	0.755 ^b^
Headache	91% of patients	AU.	64	1.859	0.393	63	1.794	0.480	0.443 ^b^
Sore throat	62% of patients	AU.	49	1.776	0.422	38	1.632	0.633	0.2573 ^b^
Cough	55% of patients	AU.	41	1.902	0.490	36	1.833	0.378	0.606 ^b^
Pain in muscles **	51% of patients	AU.	39	1.872 **	0.656	32	1.469 **	0.671	0.013 ^a^
Runny nose	27% of patients	AU.	23	1.957	0.475	15	1.733	0.458	0.247 ^b^
Loss of smell **	16% of patients	AU.	8	2.375 **	0.744	14	1.714 **	0.469	0.036 ^b^
Loss of taste	4% of patients	AU.	4	2.500	0.577	2	3.000	0	0.312 ^b^
Physical activity **		AU.	68	12.75 **	2.984	72	13.85 **	3.005	0.019 ^b^
Physical activity (daily walk) **		min	68	7.279 **	8.614	72	12.57 **	13.45	0.008 ^b^
Decreased attention (d2-test)		%E (errors)	68 68	28.34	21.47	72	26.00	26.83	0.189 ^b^
URTI **		WI score	68	17.59 *	6.497	72	14.69 *	5.832	0.006 ^a^
QOL		WI score	68	36.29	12.15	72	37.14	12.81	0.163 ^b^
Blood serum IL-6 (normal level < 7 pg/mL)		pg/mL	68	12.60	54.29	72	9.39	17.86	0.970 ^b^
D-dimer (normal range from 0.1 to 0.5 mg/L)		mg/L	68	0.812	1.528	72	4.431	32.94	0.672 ^b^
C-reactive protein (normal level < 5 mg/L)		mg/L	68	12.74	13.82	72	16.86	23.30	0.989
ALT (normal level < 35 U/L)		U/L	68	27.69	19.90	72	26.79	20.46	0.831
AST (normal level < 32 U/L)		U/L	68	25.32	19.07	72	26.83	19.49	0.241
Total WBC count (normal range: 3.6–11.0 × 10^9^ cells/L)		10^9^/L	68	5.872	1.863	72	5.271	1.936	0.064
Erythrocytes, RBC (normal range: 3.8–5.8 × 10^12^ cells/L)		10^12^/L	68	4.699	0.479	72	4.770	0.601	0.231
Hemoglobin, Hb (normal range: 13.5–17.0 g/dL)		g/dL	68	12.95	1.647	72	13.51	1.706	0.053
Hematocrit, HCT (normal range: 40–50, L/L)		L/L	68	40.48	4.924	72	41.56	6.179	0.064
Platelet count (normal range: 150–380 × 10^3^ cells/μL)		10^3^ μL	68	207.1	49.49	72	200.8	51.22	0.565
Neutrophil count (normal range: 1.8–7.5 × 10^9^ cells/L)		10^9^/L	68	60.090 *	11.80	72	62.49 *	13.43	0.458
Lymphocyte count (normal range: 1.0–4.0 × 10^9^ cells/L)		10^9^/L	68	28.72 *	11.47	72	27.54 *	12.33	0.560
Monocyte count (normal range: 0.1–1.0 × 10^9^ cells/L)		10^9^/L	68	9.194 *	15.47	72	6.457 *	3.572	0.132
Eosinophil count (normal range: 0.1–0.4 × 10^9^ cells/L)		10^9^/L	68	1.615 *	1.391	72	1.285 *	1.136	0.144
Basophil Count (normal range: 0.01–0.1 × 10^9^ cells/L)		10^9^/L	68	0.471 *	0.229	72	0.477 *	0.296	0.722

*—over the normal range; **—significantly different from a parallel group; ^a^—unpaired parametric *t*-test; ^b^—unpaired nonparametric Mann–Whitney rank test.

**Table 2 pharmaceuticals-16-01196-t002:** Schedule of examinations and procedures.

	Treatment							Follow-Up
	Day 1 Screening	Day 3	Day 5	Day 7	Day 9	Day 11	Day 14	Day 21
Doctor’s visits	1 Baseline			2			3	4
Eligibility check/information	*							
Informed consent	*							
Clinical examination	*			*			*	*
Enrollment and allocation to intervention	*							
Treatment (Kan Jang or placebo)	*	*	*	*	*	*	*	
*Biomarker assessments*								
Body temperature (fever)	*	*	*	*	*	*	*	*
COVID-19 PCR test	*			*			*	*
Blood serum cytokine IL-6 (pg/mL)	*	*		*			*	
D-dimer (mg/L)	*			*			*	
C-reactive protein (mg/L)	*			*			*	
Blood cell count analysis	*			*			*	
ALT/AST	*			*				
*Clinician- and observer-reported outcome assessments*								
Cognitive performance (tests for attention and memory): d2 test Wisconsin URS survey score	* *	*		* *			* *	*
Drug intake accountability							*	
Adverse events				*			*	*
*Patient-reported outcome assessments*								
Mild COVID-19 symptoms: Fatigue Headache Loss of smell Loss of taste Rhinorrhea (nasal discharge) Cough Pain in muscles Sore throat	*	*	*	*	*	*	*	*
Workout, min	*			*			*	*
Physical activity (questionnaire)	*			*			*	*
Paracetamol intake recording Rescue medication intake recording	* *	* *	* *	* *	* *	* *	* *	

## Data Availability

Data are contained within the article and Appendix A.

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
