# Peer review of "Efficacy of Kan Jang® in Patients with Mild COVID-19: A Randomized, Quadruple-Blind, Placebo-Controlled Trial"

_pharmaceuticals, 2023, doi:10.3390/ph16091196_

Round 1
Reviewer 1 Report
1. Check the abbreviations throughout the manuscript and introduce the abbreviation when the full word appears the first time in the abstract and the remaining for the text and then use only the abbreviation (For example, SARS-CoV-2, IL-6, intention to treat (ITT), etc.,). Make a word abbreviated in the article that is repeated at least three times in the text, not all words to be abbreviated. Avoid the usage of abbreviation in the keywords.
2. The full form of the species should be given when the first time appears in both the abstract and in the remaining part of the manuscript and it should be followed by only the first letter of the genus (For example, Andrographis paniculata when the first time appear and followed by A. paniculata).
3. The introduction part appears less informative about the COVID-19, thus this section should be indicated as detailed to understand the manuscript in clear.
4. The authors may cite recent prevalence or incidence data about COVID-19 and it should be at-least of 2022 or 2023.
5. The table and figure legends should be improved and a proper footnote should be given. All legends should have enough description for a reader to understand the table and figures without having to refer back to the main text of the manuscript. For example, the necessary abbreviations should be given.
6. In results, under the heading “2.3. Safety”, the authors should include what are all the parameters taken into account to prove the safety of the drug chosen for better understanding.
7. The technical terms (Latin Phrase) “in vitro”; “ex vivo” should be italic and it should be checked all over the manuscript.
8. In materials and methods section, the authors should include what are all the criteria followed for the selection of placebo samples since it is not mentioned clearly.
9. The conclusion seems in general. All conclusions must be convincing statements on what was found to be novel, impact based on the strong support of the data/results/discussion. Moreover, the authors may also be included the limitation of the present findings for a better understanding of the manuscript.
1. The English need improvement since there are some grammatical and syntax errors in the manuscript. For example, the words “the color-coded” may be as “color-coded”; “patients admitted” as “patients were admitted”; “to placebo” as “to the placebo”; “while rhinorrhea” as “rhinorrhea”; “Cytoprotective” as “The cytoprotective”; “described in” as “described”. The grammar mistakes which are not mentioned here are also to be checked and corrected properly.
2. There are some typing mistakes as well, and authors are advised to carefully proof-read the text. For example, the words “diagrams showing” may be as “diagrams shows”; “follow up” as “follow-up”. The typos not mentioned here are also to be checked and corrected properly.
Reviewer 2 Report
The authors describe a well-designed RCT testing a novel herbal medicine in treating COVID-19 patients with mild-moderate symptoms. The findings are of likely interest to health care professionals.
The comments are mostly minor, but compliance is an item that requires description in the Results.
Abstract: "The rate of resolution of inflammatory symptoms in the Kan Jang® group was significantly higher compared with the placebo group, and relief of the severity of cough, sore throat/pain, runny nose, and muscle soreness." Sentence needs correction.
Intro: "The latest wave was evolving several SARS-Cov-2 variants characterized by a remarkable speed, lesser severe and intense, less comorbidity, and lower death rate, but a younger population, newer symptoms like gastrointestinal, more cases with breathlessness, and
much higher positivity rate as compared to the 1st wave." Needs correction.
Figure 2 is difficult to read. Please enlarge.
"The groups did not show differences in baseline demographic, physical, and other critical clinical measurements, Table 1, except for lower physical activity scores in the Kan Jang group compared with placebo, higher muscle soreness score, loss of smell score, and upper respiratory symptoms score assessed by Wisconsin URS Survey." Needs correction.
The Results section contains descriptions that belong in Methods: Efficacy and primary endpoints. Please place all descriptions of your methodology in Methods.
Recovery time and time to fever resolution: "from >370 C to < 380 C)" needs correction.
Discussion: "The results of this study are consistent with the previous publications where Kan Jang effectively relief sore throat..." Needs correction.
Neither publication reports compliance rates. Please state the compliance rate deemed acceptable by the investigators and the actual number of patients meeting this criteria for the treatment and placebo groups.
Moderate English editing is needed.
Round 2
Reviewer 1 Report
1. There are some grammatical, alignments and typographical errors are noted in the manuscript and it should be thoroughly checked and corrected throughout the manuscript. For example, the words “that admitted” may be as “admitted”; “with cough” as “with a cough”; “patients patients” as “patients”; “patients all” as “patients in all”; “for progression” as “for the progression”.
2. This suggestion is not properly carried out. Check the abbreviations throughout the manuscript and introduce the abbreviation when the full word appears the first time in the abstract and the remaining for the text and then use only the abbreviation. For example, both full and short forms used for intention to treat (ITT) in page 3 and 14). These type of corrections need to be checked all other abbreviations used in the manuscript.
3. The genus and species name used in the manuscript should be italic all over the manuscript. For example in page 2, “Andrographis paniculata” is not given in italic form. Similarly for “Eleutherococcus senticosus” in the same page.
4. The authors should improve the quality of the images used in the manuscript with high resolution for better understanding. For example, in figure 3 the words mentioned in not able to see properly since it is blurred and it should be rectified and also the background colour is not able the see properly. The figure legends is not also properly visible hence, it should be rectified.
Reviewer 2 Report
The authors have addressed my concerns except for:
The revised sentence in the abstract still needs correction:
The rate of resolution of inflammatory symptoms in the Kan Jang® group was significantly higher, decreasing relief of the severity of cough, sore
throat/pain, runny nose, and muscle soreness compared with the placebo group." Suggest "...higher, marked by reduced severity of cough,...
A minor correction is needed, as state above.
